# SOI-FET Sensors with Dielectrophoretic Concentration of Viruses and Proteins

**DOI:** 10.3390/bios12110992

**Published:** 2022-11-08

**Authors:** Olga Naumova, Vladimir Generalov, Dmitry Shcherbakov, Elza Zaitseva, Yuriy Zhivodkov, Anton Kozhukhov, Alexander Latyshev, Alexander Aseev, Alexander Safatov, Galina Buryak, Anastasia Cheremiskina, Julia Merkuleva, Nadezhda Rudometova

**Affiliations:** 1Rzhanov Institute of Semiconductor Physics, Siberian Branch of the Russian Academy of Science, 630090 Novosibirsk, Russia; 2Federal State Research Institution State Research Center of Virology and Biotechnology “Vector”, 630559 Koltsovo, Russia; 3Faculty of Automation and Computer Engineering, Novosibirsk State Technical University, 630073 Novosibirsk, Russia; 4Department of Physics, Novosibirsk State University, 630090 Novosibirsk, Russia

**Keywords:** biosensor, SOI-FET, dielectrophoresis, spike protein, SARS-CoV-2

## Abstract

Quick label-free virus screening and highly sensitive analytical tools/techniques are becoming extremely important in a pandemic. In this study, we developed a biosensing device based on the silicon nanoribbon multichannel and dielectrophoretic controlled sensors functionalized with SARS-CoV-2 spike antibodies for the use as a platform for the detection and studding of properties of viruses and their protein components. Replicatively defective viral particles based on vesicular stomatitis viruses and HIV-1 were used as carrier molecules to deliver the target SARS-CoV-2 spike S-proteins to sensory elements. It was shown that fully CMOS-compatible nanoribbon sensors have the subattomolar sensitivity and dynamic range of 4 orders. Specific interaction between S-proteins and antibodies leads to the accumulation of the negative charge on the sensor surface. Nonspecific interactions of the viral particles lead to the positive charge accumulation. It was shown that dielectrophoretic controlled sensors allow to estimate the effective charge of the single virus at the sensor surface and separate it from the charge associated with the binding of target proteins with the sensor surface.

## 1. Introduction

As of November 2020, there were nearly 56 million confirmed cases and more than one million deaths due to the outbreak of coronavirus disease (COVID-19) caused by the respiratory syndrome coronavirus 2 (SARS-CoV-2) [1]. As of February 2021, there were over 107 million confirmed cases and 2.3 million deaths [2]. As of October 2022, over 615 million confirmed cases and over 6.5 million deaths have been reported globally [3]. The coronavirus outbreak has highlighted the urgent need to develop analytical tools and diagnostic devices with such set of main requirements as (1) high sensitivity and low detection limit, (2) large dynamic range, (3) high specificity and high selectivity, (4) rapid and real time detection, (5) validity and good reproducibility, (6) lab-independent home-test devices, portability and (7) easy fabrication processes. Recently, intensive efforts in this field have shown that silicon-on-isolator (SOI) FET sensors are the versatile platform to detect any bioparticles including viruses, nucleic acids, and proteins [4,5,6,7,8,9]. The listed main requirements for such sensors are provided by the principle of their operation and the compatibility with the conventional CMOS technology (silicon industrial technology). Note that CMOS-compatibility and the so-called top-down fabrication process offer several advantages for SOI-based sensors over others FET-sensors, for example, based on the carbon nanotubes [10,11], graphene [12], etc. This allows one (1) to address the issues of reproducibility, scalability, mass industrial manufacturing and integration with on-chip addressing and signal processing components and (2) to use a global gating scheme in multiplexing and various calibration methods, for example, to account the small device-to-device variations and drift [4].

The device consists of a nanowire or nanoribbon with ohmic (source-drain) contacts at the ends fabricated in the top silicon layer of SOI structures. In this design, the device acts as a dual-gate field-effect transistor. The SOI buried oxide (BOX) is used as a gate-oxide and the SOI substrate is used as a back-gate (BG) to adjust the initial conductivity of the sensor due to the field effect. Due to the field effect, any particle adsorbed on the sensor surface modulates (increases or decreases) the sensor conductivity acting as a second (local virtual) gate. It offers real-time and high-sensitivity detection for any type of bioparticles.

To ensure the adsorption of, primarily, target bioparticles, the sensor surface is functionalized by specific probes, for example, with antibodies. This complicates the device manufacturing process, but provides the selectivity, specificity and increases the reliability of the analyte detection. The signal from nonspecific adsorption (background signal) is recorded using a second sensory element (without specific probes). The sensor response to specific interactions is determined in the differential mode as
(1)$$\mathrm{Resp}=\left|\frac{I_{\mathrm{ds}}^{w}-I_{\mathrm{ds}}^{r}}{I_{\mathrm{ds}}^{r}}\right|$$

Here, $I_{\mathrm{ds}}^{w}$ and $I_{\mathrm{ds}}^{r}$ are the source-drain current for working and reference sensors with probes at the surface and without probes, respectively. Thus, the device is potentiometric and requires two sensory elements. Note that most target bioparticles, including proteins and viruses, are charged molecules or strongly polar molecules. Therefore, both *I*_ds_ components in Equation (1) are determined by the effective surface charge (surface net-charge of the analyte and the mobile charge of the electrolyte in which these particles are placed). The differential response for sensors with and without ABs is used to extract the signal associated with the specific binding of target particles to the sensor surface.

Nanoribbon sensors, in comparison with nanowire ones, ensure good repeatability, low noise, easy top-down fabrication compatible with the conventional CMOS technology and they increase the total sensing area, which is significant for the detection of analytes with a low concentration [4,6]. At a low analyte concentration, the detection limit strongly depends on the probability of adsorption of target particles on the sensor surface. Multichannel nanoribbon sensors and the dielectrophoresis (DEP) technique (DEP delivery of bioparticles to the sensor surface) can be used to overcome the limitation of particle diffusion transport to sensory elements.

As known, the force acting in a non-uniform electric field of strength *E* on a dielectrically polarized spherical particle with radius *r* is determined as [13,14]
(2)F→= 2πr3εmRe[FCM] ∇|E2|=2πr3εoεmRe[εp*−εm*εp*+2εm*] ∇|E2|

Here, Re[*F_CM_*] is the real part of the Clausius–Massotti factor, the sign of which determines the acting force direction (positive or negative DEP), *ε_m_* is the absolute dielectric permittivity of the medium, *ε*_p_* and *ε*_m_* are complex dielectric permittivity of a particle and medium, respectively.

The polarized particles can move in various directions, including the attraction or repulsion from the electrode (at *ε*_p_* > *ε*_m_* or *ε*_p_* < *ε*_m_*, respectively) by changing the applied electric field. Therefore, it is very attractive to place the sensor between DEP-electrodes to concentrate the particles on the sensor surface. Note that the bioparticles can be concentrated on the sensor surface both in the case of a negative DEP (when they are repelled away from the DEP-electrodes) and a positive DEP (when they are attracted to the sensor due to the high ∇*E*^2^ values near it) [15].

The polarizability of bioparticles reflects their uniqueness. Therefore, the DEP opens up wide-ranging capabilities for manipulating the location of the analyte and background particles between the DEP-electrodes [13,14,15,16]. In addition, the design parameters of sensors with the DEP control provide a certain selectivity in the analyte indication. Indeed, according to Equation (2), the dielectrophoretic force depends on the particle size (*F*~*r*^3^) and the electric field strength (distance between electrodes, respectively). The effective DEP control of viruses requires the electrode spacing at the range of micron units and, for controlling proteins, at a submicron range [16]. It means that, in the case of sensors with DEP-controlled viruses, the DEP force acting on viruses does not affect (acts weakly) the concentration of background proteins near the sensor element. However, despite a large number of experimental and theoretical studies on the DEP-manipulation of bioparticles (see reviews [13,14,17,18]), only a few reports [15,18,19,20,21] are devoted to such a problem as the DEP delivery of an analyte to FET sensors.

This study is aimed at developing a silicon-based biosensing device functionalized with SARS-CoV-2 spike antibodies for the use as a platform for the detection of viruses and their protein components. For this, chips with a set of nanoribbon back-gate multichannel SOI-FET sensors and sensors with lateral DEP-electrodes were used. Replicative defective virus-like particles based on vesicular stomatitis virus and human immunodeficiency virus 1 (HIV-1)-bearing SARS-CoV-2 S-protein were used to deliver proteins to sensory elements.

As known, the SARS-CoV-2 virion is covered with multiple copies of the S-protein, and its receptor-binding domain (RBD) ensures the interaction of the virus with the ACE2 receptor (angiotensin converting enzyme) on the surface of the target cell and triggers the penetration of the virus into the cell [22,23]. Therefore, the S-protein is one of the main targets for the detection of viral particles of SARS-CoV-2. The choice of virus-like particles (VP) is due to the following. Firstly, VPs are not able to infect a living cell and they are safe for research. Secondly, VPs are able to include the envelope proteins of other viruses. In addition, the same sensor design (with the same DEP electrode spacing) can be used for targeted delivery of both viruses and proteins, if the viruses are used as the protein carriers. Virus-like particles were also modified with fluorescent labels for their visualization and verification of measurements.

The results showed the high sensitivity response of the sensors at the detection of viruses in aqueous solutions. The increase in the concentration of viral particles on the sensor surface was found at MHz frequencies. It was shown that specific interaction between S-proteins at the virus surface and antibodies at the sensor surface leads to the negative charge accumulation on the sensor surface. The effective charge of the DEP polarized viral particles is positive. To extract the signal from specific and nonspecific interactions (from the target proteins and viruses themselves) resulting in the different charge sign on the sensor surface, the three-element measurement scheme, including two DEP controlled sensors (without and with antibodies) and a sensor without DEP control (without antibodies), was proposed. At a low analyte concentration in the samples, the proposed approach makes it possible to estimate the effective charge of the single virus at the sensor surface and separate it from the charge associated with the binding of target proteins with the sensor surface.

## 2. Materials and Methods

Back-gate multi- and single n-channel SOI FET-sensors with and without DEP-lateral electrodes were used in this study. The schematic and optical images of sensors are shown in Figure 1. Sensors were made on the base of commercial SOI structures of p-type conductivity with the top silicon layer thickness of 30 nm and buried oxide (BOX) thickness of 200 nm. The concentration of the acceptors in the SOI film was 2 × 10^16^ cm^−3^. The sensor elements were SOI nanoribbons 3 μm wide (for twelve-channel sensors) or 1 μm wide (for single-channel sensors), 10 μm long, having source (S) and drain (D) contact regions at the ends. Standard RCA treatment was applied to SOI-wafers before the sensor fabrication [24]. The SOI-FET sensors were fabricated by using a top-down approach and photolithography. The main steps in the fabrication process flow is described elsewhere [6]. Lateral DEP-electrodes (G1, G2 in Figure 1c) were made of highly doped CVD polysilicon. The surface of chips with a set of different sensors was protected with a teos-oxide. In the area of the sensor elements, the protective layer had windows for the open-access of solutions with the analyte.

After the fabrication, covalent modification through silane chemistry was used, as the most common technique [4]. For this, the sensor surface was (1) activated by OH-groups, (2) salinized by 25% ethanol solution of (3-Aminopropyl) triethoxysilane (APTES, Sigma Aldrich, St. Louis, MO, USA), as a typical silane agent, and (3) functionalized by monoclonal antibodies (AB) to the S–proteins. The sensor surface at different preparation stages is schematically shown in Figure 2. This method of sensor modification was chosen due to a number of advantages: oriented antibody layer, reliable fixation of antibodies, the possibility of blocking the free biosensors surface. To ensure differential measurements, part of the sensors on the chips was without ABs. The sensor surface preparation process is described in details elsewhere [25]. The glycine (Sigma Aldrich, St. Louis, MO, USA) passivation was also used to suppress nonspecific interactions at the sensor surface [26].

All the biological material and samples used in this study were prepared in the Federal State Research Center of Virology and Biotechnology “Vector” of Rospotrebnadzor, Russia. As analytes, we used the replicative defective viral particles on the base of vesicular stomatitis viruses, a representative of the rhabdovirus family and on the base of HIV-1, a representative of the lentivirus genus. The VP surface was modified by the surface spike S proteins of the SARS-CoV-2 coronavirus (20–30 proteins/virus). The remaining virion-forming proteins of vesicular stomatitis viral particles (VSVP) were represented by the vesicular stomatitis virus proteins and genomic RNA defective in the G virus coding region. The VSVP size is about 150 nm. The remaining virion-forming proteins of the human immunodeficiency viral particles (HIVP) were represented by proteins of the HIV-1 and genomic RNA defective in the Env coding region. The size of HIVP is about 100 nm. The initial VP suspensions contained 10^10^ particles/mL (corresponding to ~2 × 10^−11^ M). For the optical visualization of VPs, the fluorescent labels were formed using fluorescein-5-isocyanate (Sigma Aldrich, St. Louis, MO, USA) [27]. To increase the DEP force acting on the HIVPs (see Equation (2)), their size was increased up to 1–2 µm by the fluorescein-5-isocyanate treatment as well. Specific monoclonal antibodies to the RBD of S protein of the SARS-CoV-2 coronavirus were obtained based on the m396 antibody interacting with SARS [28]. To increase the affinity of ABs to SARS-CoV-2, its structure was changed. Leucine in position 54 was replaced by valine.

For the study, the initial suspensions of antibodies and viruses were diluted in deionized aqua (DI). Solutions of AB:DI = 1:100 (with the low AB-concentration) or 1:10 (with the high AB-concentration) were applied for the sensor surface functionalization. The concentration of viral particles in the samples was varied in the range of 10^−18^–10^−12^ M. To provide the simultaneous detection of viruses by sensors without ABs and with ABs, 1 μL of the virus-containing sample was dripped on the 3 mm^2^ chip surface with the set of sensors.

Electric measurements were made using Multifunction DAQ NI USB-6363 (USA). During all virus detection measurements, the reference electrode (RE) placed into the solution was grounded (Figure 1a), the constant voltage *V_ds_* = 0.15 V was applied between the source and drain. SOI substrate was used as the back-gate to set the subthreshold mode and provide the highest signal response for back-gated SOI-FET sensors [29].

## 3. Results and Discussion

Figure 3 shows the *I_ds_*(*t*) dependences during the VSVP detection for multichannel sensors functionalized with different concentrations of antibodies. Hereinafter, the current values for sensors in DI (the initial *I_ds_* values) are given to one level. Figure 3 also shown the typical RE-gate leakage current *I_gs_* for the sensors. The sensor current tends to decrease with the increase in the virus concentration in samples. The *I_ds_* values for AB-coated sensors are lower than for reference sensors ($I_{\mathrm{ds}}^{w}$ < $I_{\mathrm{ds}}^{r}$) over the entire range of virus concentrations. However, the behavior of $I_{\mathrm{ds}}^{w}\left(t\right)$ can be non-monotonic. For sensors with functionalization AB:DI = 1:100 (with the low AB-concentration at the surface), the increase in $I_{\mathrm{ds}}^{w}$ values is obviously observed with some time delays relative to the moment of adding viruses to the solution (peaks 1–3 in Figure 3b).

Figure 4 shows the response for the sensors functionalized with the different AB concentration depending on the concentration of viruses (N_VSVP_) in the samples. The error bar and the linear fits of the *Resp*(N_VSVP_) dependences are also shown in Figure 4. The error bar in Figure 4 represents the maximum value of the standard deviation of measurements obtained on three chips for multichannel sensors in the virus concentration range of 2 × (10^−18^–10^−15^) M.

The detection responses of the sensors were linear (1) in the virus concentrations ranged of 2 × (10^−18^–10^−14^) M with the linear regression of R^2^ = 0.989 for 1:10 AB-coated sensors and (2) in the range of 2 × (10^−18^–10^−15^) M with the linear regression of R^2^ = 0.989 for 1:100 AB-coated sensors. As known, the linear fits of the response concentration dependences can be used to determine the limit of detection (LOD) for sensors. The LOD can be defined as the lowest detectable molar concentration of target particles and can be calculated as Y_LOD_ = Y_blank_ + 3SD_blank_ (here Y_blank_ is mean of the calculated blank measurement and SD_blank_ is the standard deviation of the blank measurement) [30]. In this case, the calculated LOD values are 10^−18^ M and 4 × 10^−19^ M for sensors with the low and high AB concentrations on the surface, respectively (see Figure 4). The 10^−18^ M molar concentration corresponds to 0.6 virus in the 1 µL sample. Therefore, we believe that at the low analyte concentration, the experimental detection limit for sensors should be defined. If the sample contains the single target particles, the experimental LOD should be determined by the minimum value of the reliably measured sensor response, i.e., at the response value of 10%. For multichannel sensors under study, the experimental LOD value is about 2 × 10^−18^ M. Note that, for the first time, the possibility of detecting single viruses using silicon nanowire FETs functionalized with antibodies was shown in [5].

Thus, the sensors have their (1) subattomolar sensitivity level (LOD value is about 2 × 10^−18^ M) and (2) linear dependences of Resp in the N_VSVP_ range of 2 × (10^−18^–10^−14^) M for 1:10 AB-coated sensors and in the range of 2 × (10^−18^–10^−15^) M for 1:100 AB-coated sensors (i.e., dynamic range is about 3–4 orders).

As known, the negative charge accumulation on the surface of the n-channel sensors causes the decrease in the *I_ds_* current. The positive charge accumulation on the surface of the n-channel sensors causes the increase in their current. Therefore, the decrease in *I_ds_* with the increase in the virus concentration and the lower *I_ds_* values for AB-coated sensors (Figure 3) mean that the negative charge is caused by the specific interaction between ABs and S-proteins on the surface of viruses. We can also assume that the increase in *I_ds_* (indication of the positive charge by sensors) can be caused by nonspecific (1) virus- sensor interaction, for example, due to the non-uniform charge distribution at the virus surface or (2) virus-virus interaction on the sensor surface.

Figure 5 shows the *I_ds_*(*t*) dependences for different multichannel sensors during the measurement of samples containing HIVPs. The glycine passivation was not used to increase the nonspecific virus-sensor interaction. The increase in *I_ds_* values is observed for the HIVP concentration of more than 2 × 10^−14^ M (comp. Figure 3 and Figure 5).

Figure 6 shows the *I_ds_*(*t*) dependences for single-channel sensors without and with the dielectrophoretic control of HIVPs, measured at the frequency of 2 MHz. The reference (without antibody functionalization) DEP-sensor was used to highlight nonspecific interactions with the viruses, as the concentration of viruses on the sensor surface was found at MHz frequencies at the positive DEP. The surface of the sensor without DEP control was modified by ABs for the detection of specific AB—S-protein interaction. The obvious difference in the current values with the same ratio $I_{\mathrm{ds}}^{w}$ < $I_{\mathrm{ds}}^{r}$ (cp. with Figure 3) is observed even at a low virus concentration in the samples.

Figure 7 shows the optical images of the corresponding sensors after the HIVP detection in the concentration range of 2 × (10^−17^–2 × 10^−12^) M. Single HIVPs are observed on the sensor surface without DEP control, while the DEP-controlled sensor surface is completely covered by the luminous viral shell.

Thus, the obtained results confirm that the nonspecific virus-sensor interaction can leads to the positive charge accumulation on the sensor surface. In this case, there is the question of how to estimate the charges of the different sign (from specific and nonspecific interactions) with using sensors.

Figure 8 shows the dependencies *I_ds_*(*t*) for the DEP-controlled working and reference sensors measured at the HIVP concentration of 2 × 10 ^−17^ M in the sample. Additionally, the *I_ds_*(*t*) dependence for the reference sensor without DEP control is shown in Figure 8.

As expected, for DEP-controlled sensors: (1) the working sensor current is lower than that for the reference sensor, $I_{\mathrm{ds}}^{w}$ < $I_{\mathrm{ds}}^{r}$ (curve 2 and curve 1 in Figure 8, respectively) and (2) the sensor current values in DI is lower than that in HIVP-contained sample, $I_{\mathrm{ds}}^{r}$(DI) < $I_{\mathrm{ds}}^{r}$(HIVP). For reference sensors: (1) the DEP-sensor current values are higher than for the sensor without DEP control, $I_{\mathrm{ds}}^{r}$(DEP) > $I_{\mathrm{ds}}^{r}$ (curves 1 and 3 in Figure 8, respectively) and (2) practically the same current values are observed in DI and HIVP-contained sample for the sensor without DEP control, $I_{\mathrm{ds}}^{r}$(DI)~$I_{\mathrm{ds}}^{r}$(HIVP) (curve 3 in Figure 8).

At the low concentration of the target viral particles of 2 × 10^−17^ M (corresponding to 12 HIVP/µL), the obtained ratios for reference sensors mean that the dielectric permeabilities of the DI and the HIVP-sample are practically the same, and the DEP-sensor react to the single adsorption event. In this case, the response for reference sensors can be used to extract the signal, primarily associated with the change in the effective surface charge ∆Qs due to the DEP concentration (nonspecific adsorption) of virus.

According to the balance of charges, the change in the effective surface charge associated with the nonspecific adsorption of viruse ∆*Q_s_*_(*nonsp*)_ causes the equivalent change in the sensor charge. For the back-gate SOI-FET-sensor, it is the change in the charge of free carriers (in our case, electrons) ∆*Q_e_* that can be estimated as:(3)$$\Delta Q_{s\left(nonsp\right)}=\Delta Qe=C_{box}\left(\Delta V_{th}^{r}\left(DEP\right)-\Delta V_{th}^{r}\right)$$

Here, *C_box_* is the buried oxide capacitance; $\Delta V_{th}^{r}\left(DEP\right)$ and $\Delta V_{th}^{r}$ are the changes in the threshold voltages caused by the addition of the analyte to the solution for reference sensors with and without the DEP control, respectively.

As mentioned above, the differential response for sensors with and without ABs is used to highlight the signal associated with the specific binding of target particles (in our case, S-proteins) to the sensor surface. The obtained $I_{\mathrm{ds}}^{w}$ < $I_{\mathrm{ds}}^{r}$ ratio for DEP-controlled sensors means that the specific interaction between ABs and S-proteins leads to the negative charge accumulation on the sensor surface. According to the balance of charges, the change in ∆*Q_s_* caused by the specific binding of target particles can be estimated as:(4)$$\Delta Q_{s\left(sp.\right)}=\Delta Q_{e}=C_{box}\left(\Delta V_{th}^{w}\left(DEP\right)-\Delta V_{th}^{r}\left(DEP\right)\right)$$

Here, $\Delta V_{th}^{w}\left(DEP\right)$ and $\Delta V_{th}^{r}\left(DEP\right)$ are the changes in the threshold voltages for working and reference DEP-sensors, respectively.

The results showed that the detectable charge of polarized HIVP is about 2.5 × 10^−8^ C/cm^2^ in the samples under study. The charge associated with the binding of target S-proteins at the HIVP surface and ABs at the sensor surface is about −1.6 × 10^−8^ C/cm^2^.

We also note that the positive DEP for the HIVPs observed in the experiments means that, at the MHz frequencies, the dielectric permeability of viruses is more than that of DI, i.e., *ε*_HIVP_* > 78.

## 4. Conclusions

The sensors used in this study have (1) the subattomolar sensitivity level, (2) the dynamic range of about four orders for the N_VSVP_ range of 2 × (10^−18^–10^−14^) M and (3) real-time detection viruses with the target proteins on the surface.

Positive dielectrophoresis (*ε*_HIVP_* >78) was found for modified S-protein viral particles based on human immunodeficiency viruses in aqueous solutions at MHz frequencies.

The three-element scheme, including DEP-controlled sensors and sensor without DEP control, was proposed to extract the signal from the specific and nonspecific binding and estimate the effective charge associated with the binding of target proteins on the transport virus surface and the virus itself. The detectable charge associated with the non-specific adsorption of single HIVP was found to be 2.5 × 10^−8^ C/cm^2^ in the samples under study. The charge associated with the binding of target S-proteins on the virus surface to the sensor surface was found to be −1.6 × 10^−8^ C/cm^2^.

We believe that the approach used in this study, its capabilities and potential for characterizing virus-receptor interactions, such as modification of viruses by target proteins, DEP-concentration of transporter viruses on the sensory element, their visualization, estimation of charges for specific and nonspecific binding of individual virus could provide unique opportunities for fundamental virology and biomedicine, vaccine discovery, drug discovery.

## Figures and Tables

**Figure 1 biosensors-12-00992-f001:**
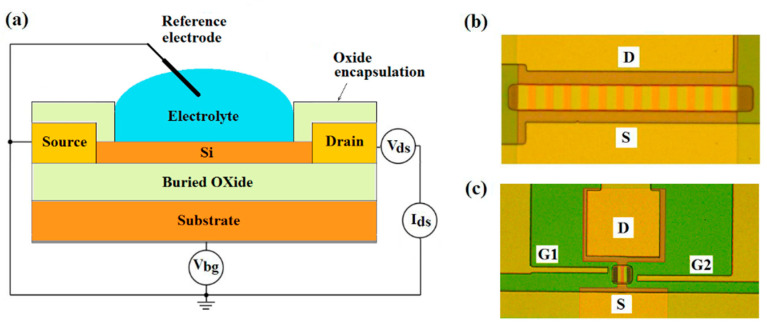
(**a**) Schematic cross-section image of a sensor. Optical images for (**b**) twelve-channel and (**c**) single-channel sensors with DEP-electrodes.

**Figure 2 biosensors-12-00992-f002:**
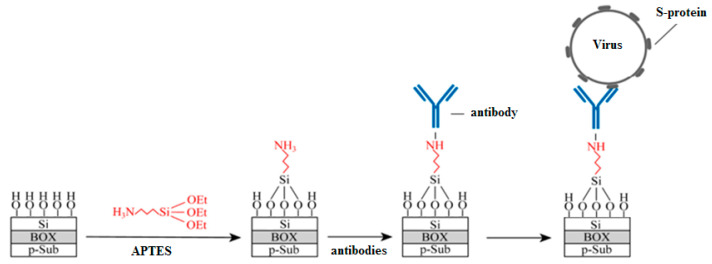
Schematic image of the sensor surface preparation.

**Figure 3 biosensors-12-00992-f003:**
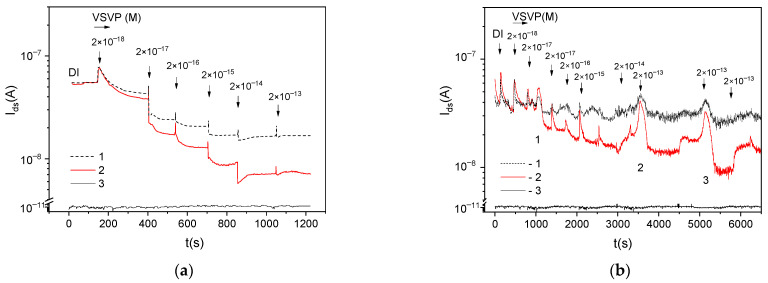
*I_ds_*(*t*) dependences during the VSVP detection for multichannel sensors (1) without ABs and (2) with ABs at the surface after the functionalization in solutions AB:DI (**a**) 1:10 and (**b**) 1:100; (3) *I_gs_*(*t*) dependences for sensors.

**Figure 4 biosensors-12-00992-f004:**
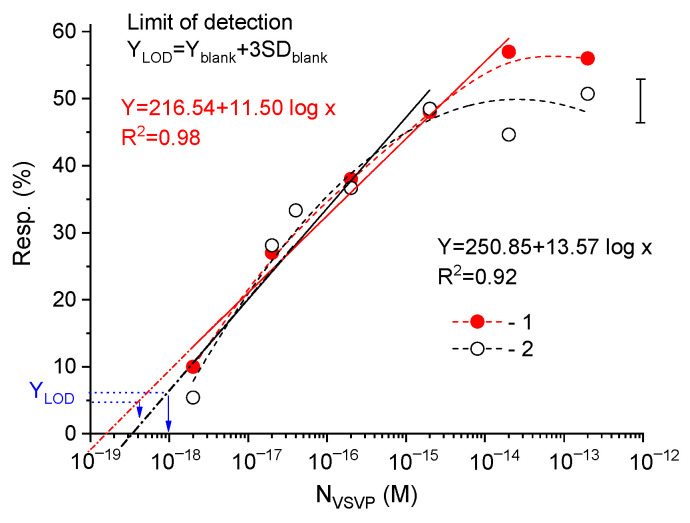
Responses of multichannel sensors functionalized by ABs in solutions (1) 1:10 and (2) 1:100 versus VSVP concentration ranging from 2 × 10^−18^ to 2 × 10^−13^ M.

**Figure 5 biosensors-12-00992-f005:**
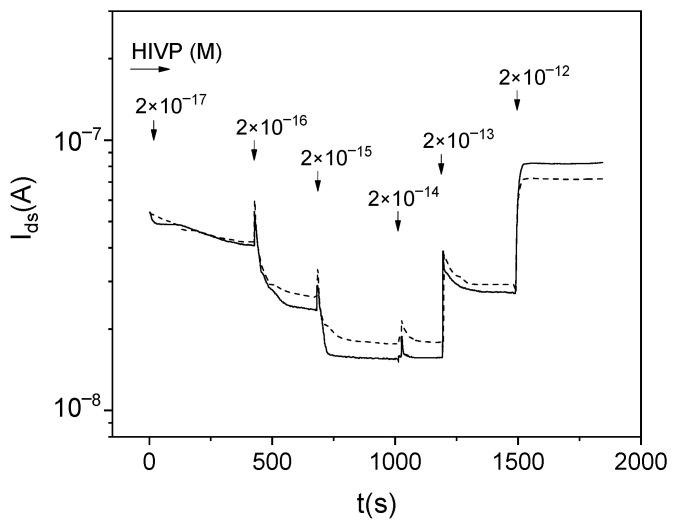
*I_ds_*(*t*) dependences during the HIVP detection for two multichannel sensors functionalized with ABs in solutions 1:10, without passivation in glycine.

**Figure 6 biosensors-12-00992-f006:**
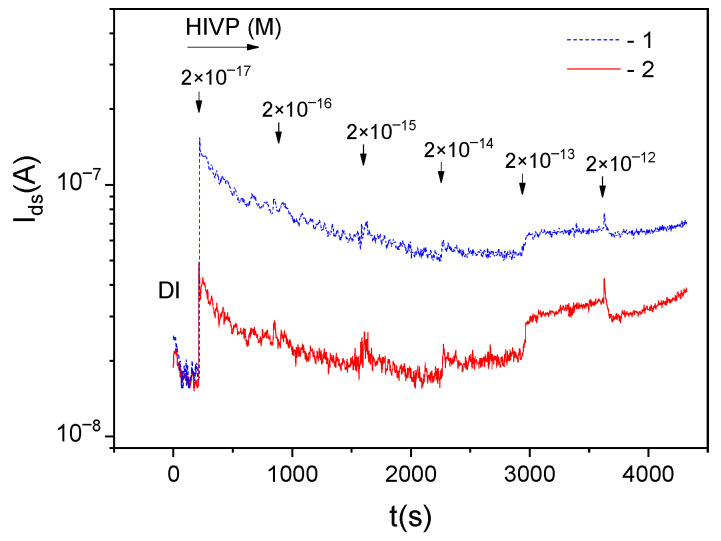
*I_ds_*(*t*) dependences during the HIVP detection for single-channel sensors (1) without ABs, with the DEP control measured at the frequency of 2 MHz and (2) with ABs, and without DEP control. The functionalization in solution 1:100, without the passivation in glycine.

**Figure 7 biosensors-12-00992-f007:**
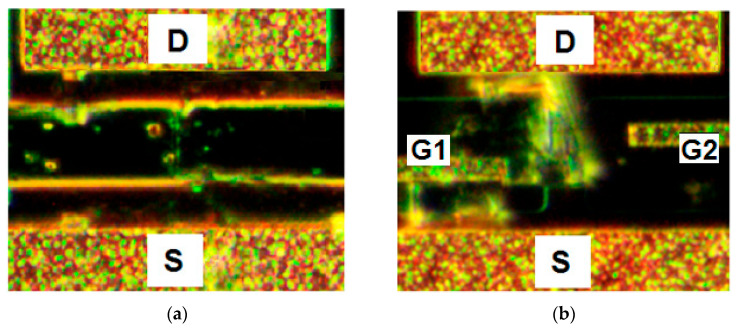
Dark field optical images for sensors (**a**) without and (**b**) with the DEP control after detecting HIVPs in the concentration range of 2 × (10^−17^–10^−12^) M at the frequency of 2 MHz. The sensor surface is (**a**) functionalized by ABs and (**b**) not functionalized by ABs.

**Figure 8 biosensors-12-00992-f008:**
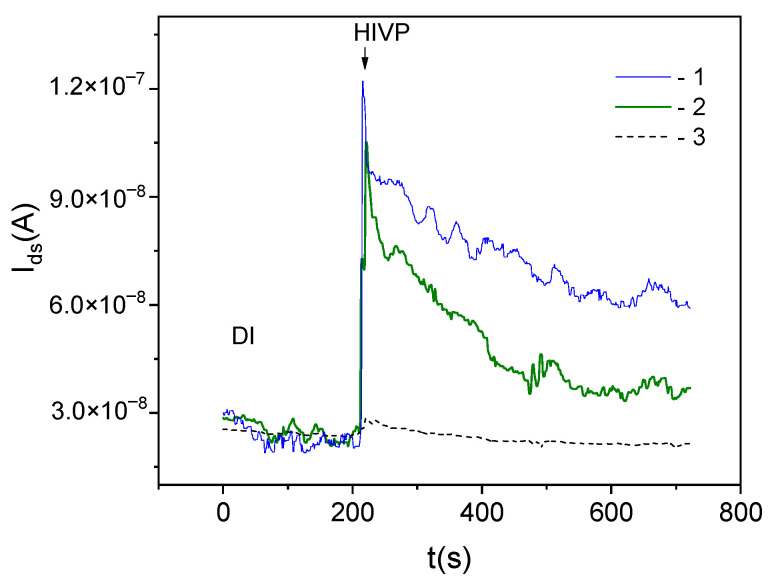
*I_ds_*(*t*) dependencies at the HIVP concentration of 2 × 10 ^−17^ M for single-channel sensors (1, 2) with the DEP control, measured at the frequency of 2 MHz, and (3) without DEP control; (1, 3) without and (2) with ABs on the surface; functionalization in solution 1:100, without the passivation by glycine.

## Data Availability

Not applicable.

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
