# Peer review of "SOI-FET Sensors with Dielectrophoretic Concentration of Viruses and Proteins"

_biosensors, 2022, doi:10.3390/bios12110992_

Round 1
Reviewer 1 Report
The work reports the usage of SOI FETs for detecting the SARS-COV-2 virus.
1) The first line in the introduction says November 2020 and now it is October 2022. Please update it with recent references. The works on COVID-19 biosensors using FET platforms (Materials Today: Proceedings 2022, 49, 2546-2549, ACS Appl. Mater. Interfaces 2021, 13, 8, 10321–10327) can be included in the introduction and the advantages or SOI FETs over other FETs has to be discussed.
2) The schematic in Figure 1 must be redrawn. The electrolyte, reference electrodes and the wirings have to be indicated clearly
3) The concentrations at 10-18 M are a bit suspicious to me as only 1uL of the samples were dripped (Effectively there will be 0.6 viral samples on a 3mm^2 sensing surface). Only one molecule producing the sensing signal on two sensors?
4) I am interested to see the reproducibility of the sensors. The error bars in Figure 4 along with the number of sensors tested.
5) I also suggested taking the baseline measurements on a device and adding them to the main manuscript.
6) The last paragraph in the conclusions has to be rewritten without brackets
7) Language check is required. Please avoid terms like " Fast analysis speed",
8) Please add a photograph of the whole device instead of a cropped image in Figure 1(c)
Author Response
Dear Reviewer,
Thank You very much for comments and suggestions.
According Your recommendations, the following changes were made.
- Introduction, Page 1, first paragraph: the coronavirus disease data for October 2022 was inserted.
Introduction, Page 2, line 47: it was added “Note that CMOS-compatibility and the so-called top-down fabrication process offer several advantages for SOI-based sensors over others FET-sensors … “
References: references [10-12] on the works on COVID-19 biosensors using FET platforms on the base CNT and graphene were included
- The schematic in Figure 1 was redrawn.
- The obvious response is observed for the virus concentration ≥ 2x10-18 M for multichannel sensors (Figure 4). This is ≥ 1.2 viruses in 1 µL-sample. However, this is only an estimation. We can’t control how many viruses there are really in drop of sample. Even for 10-18 M concentration, it may be randomly several viruses and zero viruses.
In the Results and Discussion section, after Figure 4: it was added
“The detection responses of the sensors were linear …. Therefore, we believe that at the low analyte concentration, the experimental detection limit for sensors should be defined. If the sample contains the single target particles, the experimental LOD should be determined by the minimum value of the reliably measured sensor response, i.e. at the response value of 10%. For multichannel sensors under study, the experimental LOD value is about 2x10-18 M”.
- Figure 4: The error bar was shown.
In the Results and Discussion section, line 224: it was added “The error bar in Figure 4 represents the maximum value of the standard deviation of measurements obtained on three chips for multichannel sensors in the virus concentration range of 2×(10-18-10-15) M.”
Figure 5 also shows the reproducibility of the sensors.
- During virus detection, we compare the current of the sensor with antibodies on the surface to the one of the sensor without antibodies on the surface . Therefore, in the Figures, the curve can be considered as a baseline.
- The last paragraph in the conclusions was rewritten without brackets
- Language was checked. " Fast analysis speed" was replaced on “rapid and real time detection”
- Photograph in Figure 1(c) was replaced
Reviewer 2 Report
I recommend its publication with minor revision and re-review as listed below.
1. Figure 1. Sensors were made on the base of commercial SOI structures of p-type conductivity- did you did the EIS study, how you justify the statement
2. Before the fabrication any pretreatment needed or else
3. Please justify about the modification process
4. Dynamic range is about 3-4 orders, usually we can observe 2 orders, please justify
5. Abstract needs more qualitative information.
6. What are the main features of the prepared sensor?
7. Relative error bars are missing in the plots; author should incorporate in the manuscript.
8. All presented equations lack confidence intervals for the slope and the intercept.
9. Author should recheck all the abbreviations, figures, and tables and their captions and calculated results.
10. Information related to the used method and materials is less in the introduction part, the author should improve with more information.
11. Quality of figures is low.
12. What is the sensitivity of the prepared sensor?
13. The place of procurement of some equipment's is missing, Author must be insert in the instrumentation section.
Author Response
Dear Reviewer,
Thank You very much for Your comments and suggestions.
- Information about conductivity was in the SOI-wafer specification from the supplier. Since the conductivity (mode of operation) of the sensor is tuned by the voltage on the substrate (back-gate), the doping accuracy is not very important.
- Materials and Methods, after Figure 1: it was added “Standard RCA treatment was applied to SOI-wafer before the sensor fabrication”.
References: reference [24] was added
- Materials and Methods, page 4, 2d paragraph: it was added
“After the fabrication, covalent modification through silane chemistry was used, as the most common technique…”
- The dynamic range depends on many factors (on the modification/functionalization/passivation of the sensor surface, the target particles themselves, etc.). So, in the report Nanomedicine 2016, 11, 2073–2082. [DOI:2217/nnm-2016-0071] the dynamic range of 5 orders was achieved at the short RNA detection.
- Abstract was changed
- The main features of the prepared sensor are 1) the target delivery of analyte to the sensor, 2) high sensitivity nanoribbon sensors, 3) compatibility with CMOS technology, and 4) ability to estimate/extract the effective charges of different sign on the target particle surface.
- Figure 4: The error bar was shown.
In the Results and Discussion section, line 224: it was added “The error bar in Figure 4 represents the maximum value of the standard deviation of measurements obtained on three chips for multichannel sensors in the virus concentration range of 2×(10-18-10-15) M.”
- In this study, only the equation (1) we use to calculate the dose dependence for the sensor response during the virus detection. The scale bare, the slopes and the intercepts for Resp(NVSVP) dependences are shown in Figure 4.
- All was rechecked.The LVP abbreviation was replaced by HIVP
- Introduction, page 3, line 116: it was added “Replicative defective virus-likes particles based on vesicular stomatitis virus and human immunodeficiency virus 1 (HIV-1) ….
- Quality of figures was improved.
- In the Results and Discussion section, after Figure 4: it was added
“ The detection responses of the sensors were linear ….. For multichannel sensors under study, the experimental LOD value is about 2x10-18 M”.
- In Section Materials and Methods,
- lines 168, 177, 191 it was added: “Sigma Aldrich, USA”
- line 205 it was added “…USA”
Round 2
Reviewer 1 Report
I am satisfied with the changes made.
Also, I would like to see the variation of gate leakage current (Igs) during the sensing (along with figure 3).
Author Response
Dear Reviewer,
According Your recommendation:
- Figure 3 was replaced. The Igs(t) dependencies were added.
- Results and Discussion 1-st paragraph: it was added “Figure 3 also shown the typical RE-gate leakage current Igs for the sensors.”
- Figure 3 caption: it was added “… (3) Igs(t) dependences for sensors.”